# Fenclorim Increasing Butachlor Selectivity between Wheat and *Roegneria kamoji* by Seed Soaking

Wei Tang [1], Jinqiu Sun [1], Xiaoyue Yu [1], Fengyan Zhou [2], Shengnan Liu [3], Mengjie Liu [1], Yongliang Lu [1] and Yongjie Yang [1,*]

1 State Key Laboratory of Rice Biology, China National Rice Research Institute, Hangzhou 311401, China
2 Institute of Plant Protection and Agro-Products Safety, Anhui Academy of Agricultural Sciences, Hefei 230031, China
3 Institute of Plant Protection, Sichuan Academy of Agricultural Sciences, Chengdu 610066, China
* Correspondence: yangyongjie@caas.cn; Tel.: +86-571-63370582

**Abstract:** *Roegneria kamoji* Ohwi (Poaceae), a wild relative plant of wheat which is widely distributed across China, has become a dominant and problematic weed in wheat fields in some regions. We have previously confirmed that *R. kamoji* is highly tolerant to foliar-applied acetyl-CoA carboxylase (ACCase) and acetolactate synthase (ALS) inhibitors in wheat (*Triticum aestivum* L.). The sensitivity of *R. kamoji* to pre-emergence (PRE) herbicides and the basis of fenclorim increase selectivity to butachlor between wheat and *R. kamoji* were evaluated in this study. Screenhouse bioassay showed that *R. kamoji* exhibited similar sensitivity to wheat to PRE herbicides at their recommended field doses (RFD); it also showed that buatchlor provides the highest relative control for *R. kamoji* (53.4% emergence and 81.5% fresh weight reduction, respectively), while it had no impact on seedling emergence of wheat among the six PRE herbicides. When butachlor was applied at four-fold RFD, no *R. kamoji* seedlings emerged; however, it significantly reduced the above-ground biomass of wheat compared with the non-treated control. Pre-treatment with herbicide safener fenclorim by seed soaking increased the $ED_{10}$ value of butachlor to wheat from 221.8 to 1600.1 g a.i. ha$^{-1}$, thus increasing the selectivity index from 9.6 to 68.9 between wheat and *R. kamoji*. The activities of α-amylase activity and protein content during germination, and glutathione-S-transferase (GST) and β-ketoacyl-CoA synthase (KCS) in the seedlings, could be induced by butachlor in both wheat seeds with or without fenclorim pre-soaking. These results suggested that butachlor provides the highest control for *R. kamoji* and did not affect germination and emergence in wheat. The basis of fenclorim-increased selectivity to butachlor was associated with the induced GST and KCS-mediated enhanced herbicide metabolism in wheat.

**Keywords:** *Roegneria kamoji*; wheat; herbicide safener; fenclorim; selectivity



## 1. Introduction

*Roegneria kamoji* Ohwi is a perennial grass weed in the genus *Roegneria* of Triticeae (Poaceae) which is widely distributed in China, Korea, and Japan. It is usually found in uncultivated areas and field borders, and is considered a sporadic weed in cropping systems [1,2]. *R. kamoji* has been found to be highly tolerant to most post-emergence sprayed herbicides in wheat, including acetyl-CoA carboxylases (EC 6.4.1.2, ACCase) inhibitors (e.g., fenoxaprop-ethyl) and acetolactate synthase (EC 4.1.3.18, ALS) inhibitors (e.g., metsulfuron-methyl). The tolerance mechanism of *R. kamoji* to ACCase and ALS inhibitors was conferred by non-target–site mutation and GST/P450s-involved enhanced metabolism [3,4]. As a result, *R. kamoji* has become a dominant and problematic weed in wheat fields in the Middle-Lower Yangtze River region in recent years [5]. It is highly competitive with wheat and hard to eradicate due to its sizeable shoots and perennial roots (Figure S1). Once failed control by POST herbicides was confirmed, evaluation of the efficacy of commonly available pre-emergence (PRE) herbicides as alternative options is therefore needed to control *R. kamoji* in wheat.

Butachlor [2-chloro-2′,6′-diethyl-N-(butoxymethyl)-acetanilide] is a selective chloroacetanalide herbicide which is mainly labeled for pre-emergence control of annual grasses and some broad-leaved weeds in rice. It can also be used in some other cereal and vegetable fields, such as wheat, maize, cabbage, etc. [6–8]. Butachlor is absorbed primarily through germinating shoots and secondarily by roots, and is translocated throughout the whole plant, causing failure of germination and inhibiting seedling growth in weeds [9,10]. The action target and phytotoxicity mechanism of butachlor have not been evaluated extensively. Early investigations have indicated that butachlor inhibits the activity of elongase responsible for the elongation of very long-chain fatty acids (VLCFAs) and the geranyl-geranyl pyrophosphate cyclization enzymes [11–13]. The β-ketoacyl-CoA synthase (KCS), which catalyzes the condensation of malonyl-CoA with long-chain acyl-CoA, is the key rate-limiting enzyme in the initial step of the microsomal fatty acyl-CoA elongation pathway responsible for the formation of VLCFAs [14–16]. It has been deduced that VLCFA elongases (VLCFAEs) were the initial target of chloroacetanalide herbicides.

The group of chloroacetanalide herbicides, which are applied during the germination to early seedling stages, can cause crop injury as a side effect of their excellent broad spectrum of weed control. The herbicide safener fenclorim [4,6-Dichloro-2-phenylpyrimidine], which reduced the injury to rice (*Oryza sativa* L.) via induce enhancement of the glutathione-mediated merabolism of pretilachlor, was widely co-formulated in a variety of amide herbicides [17,18]. Previous studies found that fenclorim upregulated genes related to detoxification, including P450, GST, and GT, thus reducing lipid peroxidation and oxidative damage induced by pretilachlor in rice [19]. Soaking seeds with fenclorim solution before seeding and herbicide spraying showed that fenclorim could reduce the phytotoxicity of pretilachlor to rice [20–22]; however, $\alpha$-amylase activity and contents of protein were inhibited. It is also reported that the butachlor combined with 10% fenclorim increased the $ED_{10}$s of germinated seeds and two-leaf seedlings of rice by 2.3- and 2.5-fold more than butachlor EC applied alone, respectively [17]. To date, a detailed study of the effect of fenclorim reducing phytotoxicity of butachlor in wheat has not been reported.

Therefore, the objectives of this study were to determine the response of wheat and *R. kamoji* to six commonly available PRE herbicides labeled in wheat, and evaluate the effect of fenclorim on the activities of key enzymes involved in reducing the phytotoxicity of butachlor to wheat by seed soaking.

## 2. Materials and Methods

### 2.1. Plant Material and Growth Conditions

Seeds of a *R. kamoji* population ZJHZ with a high level of tolerance to ACCase and ALS inhibitors were, first, collected in an uncultivated area in Hangzhou, Zhejiang Province, China in 2019. The rhizome of this population was separately cultivated in a greenhouse at China National Rice Research Institute (CNRRI, 30.04′ N, 119.55′ E), and seeds collected in 2020 were used in this study. Details of this population can be found in our previous studies [3,4]. Wheat seeds of the variety Yangmai 25 were purchased from China National Seed Group Co. Ltd. (Beijing, China).

Seeds for all experiments were sterilized in 3% sodium hypochlorite for 5 min, washed in distilled water three times, and then soaked in distilled water or fenclorim solution for 6 h. The soaked seeds were germinated in plastic trays (28 cm × 18 cm × 7.5 cm) containing moistened filter paper at fluctuating day/light temperatures of 25/15 °C with 14 h light coinciding with the high temperature period. The germinating seeds (with 2 mm emerged radicles) were sown on the surface in plastic pots (9 cm diameter, 9.5 cm height) containing potting soil (Hangzhou Jin Hai Agriculture Co., Ltd., Hangzhou, China). The pots were covered with a thin layer of soil, placed in a screenhouse (a 6 m × 40 m chamber framed with 2 cm iron mesh and covered overhead with a transparent plastic cover to prevent rain damage, about 25/15 °C, natural light) at CNRRI, and watered as required to maintain soil moisture. There were 10 uniform germinated seeds in each pot for herbicide spraying. All

experiments were conducted in a randomized complete block design with four replications and each experiment was conducted twice (two runs).

### 2.2. Effect of PRE Herbicides on Seedling Emergence and Growth of Wheat and R. kamoji

Screenhouse experiments were conducted to evaluate the response of wheat and *R. kamoji* to PRE herbicides labeled for application in wheat for grass monocot weed control. Herbicides at their recommended field doses (RFD) were sprayed immediately after sowing using a laboratory sprayer (3WP-2000, Nanjing Institute of Agricultural Mechanization Ministry of Agriculture, Nanjing, China) equipped with a flat-fan nozzle (TP6501E) to deliver 200 L ha$^{-1}$ at 230 kPa. Details of the PRE herbicides are listed in Table 1. There was an untreated control for both *R. kamoji* and wheat, respectively. The pots were returned back to the screenhouse after spraying. Three weeks after treatment, plant emergence rate and above-ground fresh weight were determined and data were converted to a percentage of the untreated control.

**Table 1.** Detailed information of PRE herbicides used in this study.

| Herbicide | Formulation and Manufacturer | Recommended Field Dose (g a.i. ha$^{-1}$) |
|---|---|---|
| Acetochlor [2-Chloro-N-(ethoxymethyl)-N-(2-ethyl-6-methylphenyl) acetamide] | 900 g/L EC, Jiangshan Agrochemical & Chemicals Co., Ltd., Nantong, China | 900 |
| Butachlor [2-chloro-2′,6′-diethyl-N-(butoxymethyl)-acetanilide] | 60% EC, Nanshen Plant Protection Science and Development Co., Ltd., Nantong, China | 1350 |
| Chlorotoluron [3-(3-chloro-4-methylphenyl)-1,1-dimethylurea] | 25%WP, Jiangsu Kuaida Agrochemical Co., Ltd., Nantong, China | 125 |
| Diflufenican [N-(2,4-difluorophenyl)-2-[3-(trifluoromethyl)phenoxy]pyridine-3-carboxamide] | 50%WP, Jiangsu Huifeng Agrochemical Co., Ltd., Yancheng, China | 225 |
| Flufenacet [N-(4-fluorophenyl)-N-propan-2-yl-2-[[5-(trifluoromethyl)-1,3,4-thiadiazol-2-yl]oxy]acetamide] | 41%FS, Max (Rudong) Chemicals Co., Ltd., Nantong, China | 90 |
| Isoproturon [1,1-dimethyl-3-(4-propan-2-ylphenyl)urea] | 50%WP, Jiangsu Kuaida Agrochemical Co., Ltd., Nantong, China | 937.5 |

### 2.3. Effect of Fenclorim on Butachlor Activity to Wheat

To investigate whether fenclorim reduced phytotoxicity of butachlor to wheat, seeds were soaked in 0 or 30 mg L$^{-1}$ fenclorim solution for 6 h prior to germination in plastic trays. The solution was prepared by dissolving fenclorim (98%, Shanghai Aladdin Biochemical Technology Co., Ltd., Shanghai, China) in acetone and diluted with water containing 0.1% Tween 80. Ten uniform germinated seeds were planted as described above and sprayed with butachlor immediately after planting. Specifically, wheat seeds were sprayed at a dose of 0, 1/32-, 1/16-, 1/8-, $\frac{1}{4}$-, $\frac{1}{2}$-, 1-, 2-, 4-, and 8-fold RFD of butachlor, while *R. kamoji* seeds were sprayed at a dose of 0, 1/128-, 1/64-, 1/32-, 1/16-, 1/8-, $\frac{1}{4}$-, $\frac{1}{2}$-, 1-, and 2-fold RFD of butachlor. Plants were returned back to the screenhouse and the pots were arranged in a randomized complete block design. At 21 days after treatment (DAT), the number of emerged seedlings and above-ground fresh weight per pot were recorded. Four pot replicates were used for each herbicide treatment, and the experiment was conducted twice under similar conditions.

*2.4. Protein Content and α-Amylase, GST, KCS Activities Assay*

Germinated seeds were planted as described above and sprayed with $\frac{1}{2}$-, 1-, 2-, and 4-fold RFD of butachlor. To determine the effect of fenclorim on butachlor inhibition towards α-amylase activity and protein content during the seed germination process, germinated seeds from fenclorim and water-soaked seeds were carefully removed from the soil 24 h after herbicide treatment, respectively. The germ and root tissues were washed in tap water, dried with filter papers, and a fresh 2 g were collected. To determine whether the reducing phytotoxicity of butachlor to wheat is caused by the enhanced activity of key enzymes in the VLCFAs elongation pathway or metabolic enzymes, the activities of KCS and GST toward butachlor for the plants emerged from fenclorim and water soaked seeds was compared. A total of 1.5 g fresh tissue was collected at 1-leaf stage from each treatment, respectively. The collected tissue was treated with PBS prior to biochemical assays after being ground with liquid nitrogen. The 0.1 g fresh leaf sample was homogenized by 0.9 mL of PBS at pH 7.2–7.4 and centrifuged at 3500 rpm for 15 min at 4 °C. The supernatant was collected in a centrifuge tube and placed in an ice bath. The activities of α-amylase, KCS, and GST were determined by using the commercial Enzyme-Linked Immunosorbent Assay (ELISA) kits (Meimian Biotechnology Co., Ltd., Yancheng, China) according to the manufacturer's instructions. The protein content was measured by using the coomassie blue staining method [23]. The experiment was conducted twice in a completely randomized design, with four replications.

*2.5. Statistical Analysis*

The bioassay experiments in the screenhouse were conducted using a completely randomized design with four replications. The entire experiments were conducted twice (runs). Data from the two runs of the experiment were pooled because of the non-significant interaction in the two repeated experiments, and are expressed as the means ± standard error. Mean comparison was analyzed using Fisher's protected least significant difference test (SPSS software, version 13.0, Chicago, IL, USA), where the overall differences were significant ($p \leq 0.05$).

Regression analyses were conducted using ORIGIN software (Version 2022; OriginLab Corp., Northampton, MA, USA). The dose–response curves were obtained by non-linear regression using the following four-parameter log-logistic Equation (1) [24]:

$$y = C + \frac{D - C}{1 + (x/ED_{50})^b} \tag{1}$$

In this model, *C* is the lower limit, *D* is the upper limit, *b* is the relative slope of the herbicide dose resulting in 50% above-ground fresh weight reduction ($ED_{50}$), *x* is the herbicide dose, and *y* represents plant fresh weight as a percentage of the control. Based on the regression parameters, the ratio of the herbicide dose that caused a 10% crop above-ground fresh weight reduction and the dose that caused a 90% weed above-ground fresh weight reduction were calculated, and the herbicide selectivity index (*SI*) was calculated according to Equation (2) [25]:

$$SI = \frac{ED_{10(wheat)}}{ED_{90(R.kamoji)}} \tag{2}$$

## 3. Results

*3.1. Effect of PRE Herbicides on Seedling Emergence and Growth of Wheat and R. kamoji*

Spraying seeds of wheat and *R. kamoji* immediately after sowing with PRE herbicides at their RFD caused different effects on seedling emergence and growth (Table 2). Both wheat and *R. kamoji* were susceptible to acetochlor (<4% emergence and fresh weight of control). None of the *R. kamoji* seeds emerged from the application of flufenacet and isoproturon; however, these herbicides also caused strong inhibition of the growth of wheat seedlings. The application of butachlor, chlorotoluron, and diflufenican was unable to

inhibit the emergence of *R. kamoji* completely; however, they reduced the above-ground biomass significantly compared to the nontreated control. Among the three herbicides, butachlor and diflufenican showed 97% and 100% seedling emergence on wheat, respectively; however, butachlor provided greater reduction in the emergence and growth of *R. kamoji*. The results of the dose–response assay indicated that *R. kamoji* was moderately tolerant to diflufenican, with a 52% emergence rate and 42% fresh weight of control at eight-fold RFD, respectively (Figure S2). Although no visual symptoms of injury were observed, the application of butachlor caused a 20% reduction in the seedling above-ground fresh weight of wheat at 21 DAT. The growth of wheat seedlings gradually recovered and no biomass differences were detected 40 DAT (data not shown). These results indicated that buatchlor provides the highest control for *R. kamoji* while it has no impact on the seedling emergence of wheat.

**Table 2.** Effect of different PRE herbicides on seedling emergence inhibition (%) and above-ground fresh weight reduction (%) in wheat and *R. kamoji* 21 days after treatment.

| Herbicide | Seedling Emergence (% of Control) | | Above-Ground Fresh Weight (% of Control) | |
|---|---|---|---|---|
| | Wheat | *R. kamoji* | Wheat | *R. kamoji* |
| Acetochlor | 3.4 (3.4) d [1] | 0 (0) c | 0.7 (0.7) f | 0 (0) c |
| Butachlor | 96.7 (3.3) a | 47.6 (4.3) b | 79.6 (2.8) b | 18.5 (3.0) b |
| Chlorotoluron | 82.5 (6.0) b | 75.6 (9.1) a | 67.6 (5.8) c | 42.1 (9.5) a |
| Diflufenican | 100 (0) a | 86.6 (4.3) a | 91.8 (2.3) a | 50.8 (3.6) a |
| Flufenacet | 65.3 (6.9) c | 0 (0) c | 42.8 (5.3) d | 0 (0) c |
| Isoproturon | 96.7 (3.3) a | 0 (0) c | 29.6 (2.7) e | 0 (0) c |

[1] The data represent the means (standard error). Means within the same column followed by different lowercase letters are significantly different according to Fisher's protected LSD at $p \leq 0.05$.

### 3.2. Effect of Fenclorim on Butachlor Activity to Wheat

The effect of the herbicide safener fenclorim on butachlor activity in wheat by seed soaking was evaluated. The results showed that, when butachlor was applied alone, no obvious foliar injury was observed (Figure 1); however, 20% to 30% above-ground biomass reduction was detected with one- to eight-fold RFD treatments. However, under fenclorim pre-soaking, the butachlor $ED_{10}$ values increased from 221.8 to 1600.1 g a.i. ha$^{-1}$, compared with the control treatment (Figure 2). As a result, the SI values increased from 9.6 to 68.9 between wheat and *R. kamoji* (Table 3). This finding suggests that fenclorim can reduce the phytotoxicity of butachlor to wheat and increase selectivity between wheat and *R. kamoji*.

**Table 3.** Sensitivity of *R. kamoji*, wheat pre-soaking in water, and 30 mg L$^{-1}$ fenclorim solution to butachlor.

| Herbicide | $ED_{90}$ [1] (g a.i. ha$^{-1}$) | $ED_{10}$ [2] (g a.i. ha$^{-1}$) | SI [3] |
|---|---|---|---|
| *R. kamoji* | 23.2 | | |
| Wheat | | 221.8 | 9.6 |
| Wheat + fenclorim | | 1600.1 | 68.9 |

[1] $ED_{90}$ refers to the butachlor dose required to reduce above-ground fresh weight of *R. kamoji* by 90%. [2] $ED_{10}$ refers to the butachlor dose required to reduce above-ground fresh weight of wheat by 10%. [3] SI, selective index. SI was calculated as the ratio between the $ED_{10}$ value of wheat and the $ED_{90}$ value of *R. kamoji*.

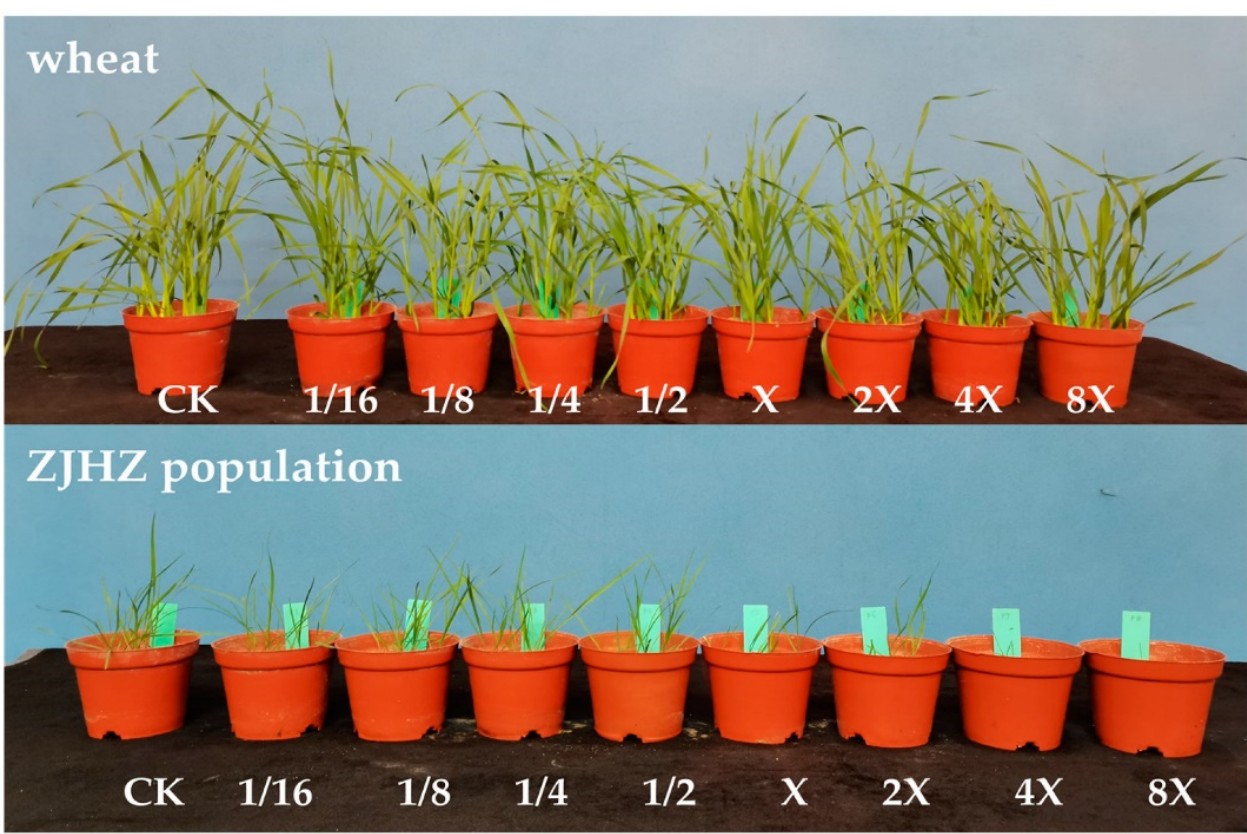

**Figure 1.** Photographs of wheat (**upper**) and *R. kamoji* ZJHZ (**lower**) 21 days after being treated with different doses of butachlor. The first row shows the untreated control (CK), followed by different doses of butachlor (X represents the recommended field dose 1350 g a.i. ha$^{-1}$).

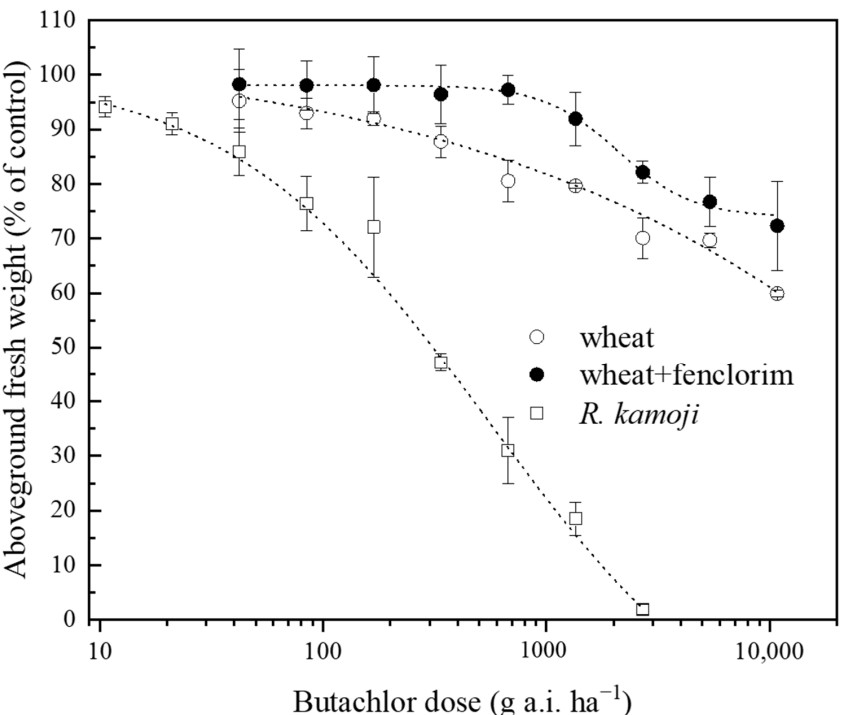

**Figure 2.** Dose–response curve for the above-ground fresh weight (% of control) of *R. kamoji* and wheat sprayed with different doses of butachlor with or without fenclorim pre-soaking. Each point represents the mean ± SE of twice-repeated experiments, each including four replications.

### 3.3. Protein Content and α-Amylase, GST, KCS Activities Assay

The results showed that, in the absence of herbicide treatment, α-amylase activity and protein content were significantly higher when seeds were pre-soaked with fenclorim over a period of 24 h (Figure 3). Furthermore, α-amylase activity increased with an increasing butachlor dose in both wheat seeds with or without fenclorim pre-soaking. Except for the seeds with four-fold butachlor treatment, there were no significant differences ($p \leq 0.05$) between wheat seeds with or without fenclorim pre-soaking under the same butachlor dose. A similar response was observed in protein content for wheat seeds which were pre-soaked with water. However, pretreatment with fenclorim showed no enhancement of protein content in wheat seeds when sprayed with a different dose of butachlor. These results indicated that α-amylase activity and protein content in wheat were stimulated when treated with butachlor, and pre-soaking with fenclorim exhibited a positive effect on seed germination.

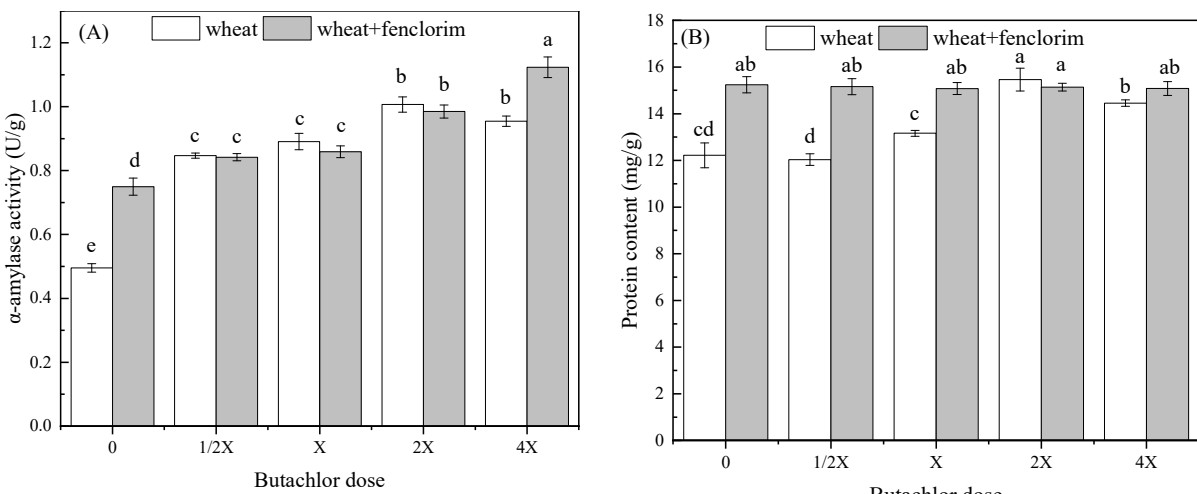

**Figure 3.** Activity of α-amylase (**A**) and protein content (**B**) in wheat seeds pre-soaked 6 h in water or in 30 mg/L fenclorim solution sprayed with different butachlor doses at 24 h after treatment. X represents the recommended field dose of butachlor 1350 g a.i. ha$^{-1}$. Each point is the mean ± SE of twice-repeated experiments, each containing four replicates. Bars designated by different lowercase letters are significantly different according to Fisher's protected LSD at $p \leq 0.05$.

The activity of GST in seedlings at 1-leaf stage wheat was significantly increased in both water and fenclorim treatments. When sprayed with 1/2-fold FRD of butachlor, the GST activities were 2.1- and 1.6-fold than that of the control treatments. At each butachlor-treated dose, the GST activity of the fenclorim treatment was higher than that of the water control (Figure 4). KCS activity was also induced by butachlor 3.4- and 3.3-fold higher than that of control; however, no difference between water and fenclorim pre-soaking treatments at the same butachlor treated dose was detected. These results indicated that both KCS and GST activities were not inhibited after the application of butachlor in wheat; instead, their activities were increased, which may have been caused by an enhanced metabolism of the herbicide. The relatively higher activity of GST in the fenclorim treatments may play a positive role in the detoxification mechanism of butachlor in wheat, thus improving crop selectivity between wheat and *R. kamoji*.

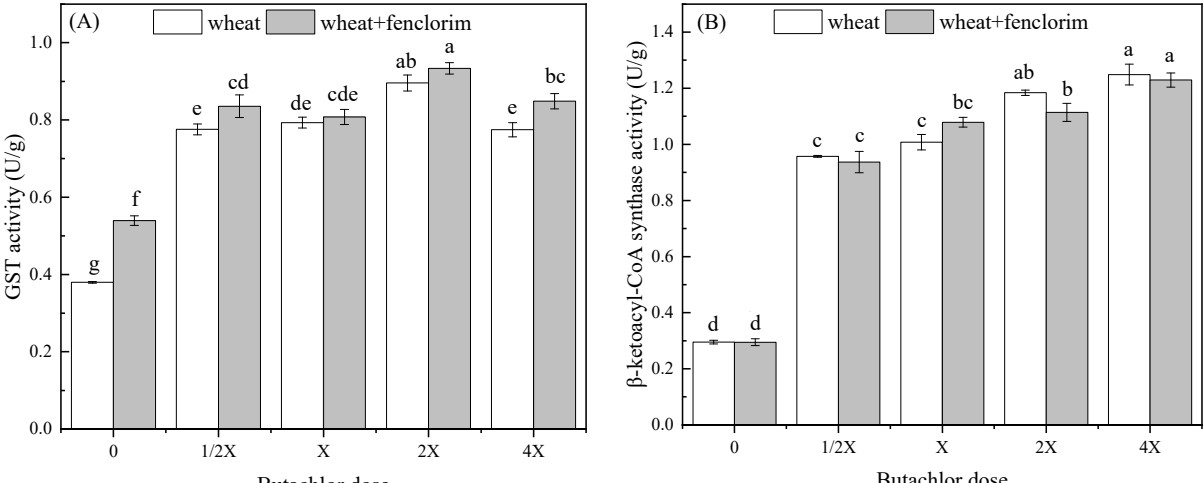

**Figure 4.** Activity of GST (**A**) and KCS (**B**) in wheat seeds pre-soaked 6 h in water or in 30 mg/L fenclorim solution sprayed with different butachlor doses at 24 h after treatment. X represents the recommended field dose of butachlor 1350 g a.i. ha$^{-1}$. Each point is the mean $\pm$ SE of twice-repeated experiments, each containing four replicates. Bars designated by different lowercase letters are significantly different according to Fisher's protected LSD at $p \leq 0.05$.

## 4. Discussion

Weeds belonging to the family Triticeae (Poaceae) are wild relative plants of wheat, and they may exhibit similar responses to various environmental stresses (e.g., pathogen infection) as wheat due to their close phylogenic relationship [26–28]. Once these weeds spread into fields, herbicide options become limited for effective management in wheat. For instance, tausch's goatgrass (*Aegilops tauschii* Coss.), an annual grass weed of the family Triticeae, is tolerant to most herbicides in wheat, and mesosulfuron-methyl is the only wheat-registered herbicide that provides effective control of this weed in China [29]. In our previous studies, *R. kamoji* showed a response pattern to all foliar-applied ACCase- and ALS-inhibiting herbicides that was similar to that shown by wheat [3,4]. The results of this study indicated that *R. kamoji* also exhibited a similar response to the PRE herbicides chlorotoluron and diflufenican, and only buatchlor provides relatively higher control for *R. kamoji* while having no impact on the seedling emergence of wheat.

Butachlor is labeled or recommended as a systemic selective PRE herbicide to be applied in wheat; however, little information is currently available regarding the phytotoxicity effects and application technology of this herbicide to wheat seedlings. It is reported that butachlor can significantly reduce the mitotic index of the dividing cells and increase the chromosomal abnormalities, which contribute to negative side effects on mitotic division in the somatic cells of wheat [10]. Petri dish tests indicated that germination percentage, length of the plumule and radicle, and seedling fresh weight were significantly decreased as the concentration of the butachlor increased [30]. The selectivity mechanisms of butachlor in rice were due to the susceptibility differences between rice and paddy weeds, as well as different absorption rates in different tissue or growth stages of the same plant [31,32]. For instance, barnyardgrass [*Echinochloa crus-galli* (L.) P. Beauv.], which is more sensitive to butachlor than rice: the $\alpha$-amylase activity and soluble sugar content during the germination process were induced after butachlor treatment in rice, while opposite results were detected in barnyardgrass [33,34].

The extension of the VLCFAs was controlled by a complex of enzymes. Besides KCS, $\beta$-hydroxyacyl-CoA dehydrogenase (HCD), $\beta$-ketoacyl-CoA reductase (KCR), and trans-2,3-enoyl-CoA reductase (ECR) were also included, and these four enzymes worked together to complete the synthesis of VLCFAs [35,36]. The mechanism of metabolic resistance/tolerance to VLCFAs inhibitors has not been elucidated extensively. Previous studies have shown the involvement of GSTs to confer tolerance to chloroacetamides in

wheat [16,35,37]. KCS catalyzing the condensation of VLCFAs is the first committed step in each elongation process. The inducement of KCS activity by fenclorim has not been reported yet. Our results suggest the possibility that multiple tolerance mechanisms involving increased KCS and GST activity confer increased selectivity between wheat and *R. kamoji*.

The chloroacetanalide herbicides are usually co-applied with safeners to protect crops from herbicide toxicity. Generally, chloroacetanalide safeners are applied via pre-soaking crop seeds in safener solution before planting or are co-applied with herbicides by spraying onto the soil at the pre-emergence stage [38,39]. However, herbicide safener may also be absorbed by weeds and accelerated their detoxification when herbicides were co-sprayed in the field. *R. kamoji* is genetically close to wheat and shares similar responses to various herbicides. which suggests that the efficacy of butachlor could be reduced via co-application with fenclorim by PRE spraying. Our results indicated that α-amylase activity and the protein content of germinated wheat seeds were increased after being sprayed with butachlor; this similar response in rice is likely to promote the germination process and confer tolerance to butachlor in wheat [19,22].

In wheat, tolerance to herbicides could be enhanced by the treatment of safeners by elevating the expression of enzymes involved in xenobiotic detoxifying, such as glutathione transferases (GSTs). To date, safeners such as benoxacor, cloquintocet-mexyl, fenchlorazole-ethyl, fenclorim, fluxofenim, mefenpyr-diethyl, and oxabetrinil have been reported to have significantly enhanced GST activity, thus exerting a range of protective and growth promoting activities in wheat [40,41]. Fenclorim is often co-formulated with pretilachlor applied in rice via upregulated genes related to detocification, including GST, and caused an acceleration of the metabolism of pretilachlor [19,42]. The results of this study support that GST activity was induced after butachlor treatment; meanwhile, activity of the key rate-limiting enzyme in the target pathway was also increased, which may have been caused by the rapid metabolism of the herbicide.

## 5. Conclusions

In conclusion, the results of this study indicate that *R. kamoji*, a wild relative plant of wheat, could not be effectively controlled by PRE herbicides chlorotoluron and diflufenican at their recommended filed doses, and that butachlor provides the highest control for this weed. The application of butachlor did not affect germination and emergence, but significantly inhibited the above ground biomass of wheat seedlings before the four-leafstage. To achieve greater selectivity to butachlor between wheat and *R. kamoji*, it is recommended to soak wheat seeds for 6 h in fenclorim solution before sowing. The basis of enhanced tolerance to buatchlor after treatment with fenclorim in wheat, including induced α-amylase activity and protein content during germination, increased GST and KCS activities in seedlings, which may play crucial roles in herbicide detoxifying.

**Supplementary Materials:** The following supporting information can be downloaded at: https://www.mdpi.com/article/10.3390/agronomy12112870/s1, Figure S1: *R. kamoji* infested wheat fields: Jingzhou, 2017 (upper); Haiyan, 2022 (lower); Figure S2: Dose response curves for the fresh weight (% of control) of wheat and ZJHZ *R. kamoji* populations treated with a range of butachlor (a), chlorotoluron (b) and diflufenican (c) doses. Each point is the mean ± SE of the two experiments each containing four replicates.

**Author Contributions:** Conceptualization, W.T. and Y.Y.; methodology, J.S, X.Y. and F.Z.; software, W.T. and S.L.; investigation, W.T., M.L. and J.S.; writing, W.T. and Y.Y.; supervision, Y.L. and Y.Y.; funding acquisition, W.T., Y.L. and Y.Y. All authors have read and agreed to the published version of the manuscript.

**Funding:** This work was financially supported by National Natural Science Foundation of China (No. 32071508), the China Agriculture Research System (CARS-01-02A), the Central Public-interest Scientific Institution Basal Research Fund (No. CPSIBRF-CNRRI-202124) and the Rice Pest Management Research Group of the Agricultural Science and Technology Innovation Program, Chinese Academy of Agricultural Sciences.

**Data Availability Statement:** Not applicable.

**Acknowledgments:** We would like to thank Yufang Chen for her valuable suggestions on preparing the manuscript drafts.

**Conflicts of Interest:** The authors declare no conflict of interest.

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
