# Peer review of "Fenclorim Increasing Butachlor Selectivity between Wheat and Roegneria kamoji by Seed Soaking"

_agronomy, doi:10.3390/agronomy12112870_

Round 1
Reviewer 1 Report
Recommendation:
Major revision
Comments to editor and authors:
Ms. Ref. No.: agronomy-2028284
Title: Fenclorim increasing butachlor selectivity between wheat and Roegneria kamoji by seed soaking
The manuscript is dealing on the selection of a good pre-emergence herbicide for Roegneria kamoji in wheat fields. Authors show that the application of butachlor and fenclorim as herbicide safener seems to be a promising option. Indeed, butachlor provides the highest control for R. kamoji and does not affect germination and emergence in wheat. Evaluation of the basis of the increased selectivity to butachlor by fenclorim was associated with a GST and KCS- enhanced metabolism in wheat.
Results of this study are interesting since they can help to weed control programs and to understand herbicide interactions with safeners to adjust management practices for maintaining wheat production.
In general, the experiments conducted are appropriate and well described in Materials and Methods, however, results are discussed poorly. To understand the real impact of butachlor and fenclorim mechanisms, I would recommend a deeper discussion. Other papers about fenclorim study additional parameters such as chlorophyll content (indeed, in some cases fenclorim increases chlorophyll content…), MDA, SOD, POD, CAT or APX. Why do authors not measure any of them? No changes are expected? Although it is generally accepted that safeners are mainly related to increasing the activities of herbicide-metabolizing enzymes, such as GSTs, P450s, GTs, and several others, something about oxidative stress should be discussed if not measured. In addition, KCS results are not discussed at all in this section. A more thoroughly review of the recently published papers in the field it is necessary to complete the bibliography and write the discussion.
Regarding to the writing, English usage in this manuscript must be substantially improved by an expert, there are many grammatical errors and sentences poorly written.
I include other comments:
-L37: include REF for tolerance of R. kamoji to foliar sprayed herbicides in wheat.
-L39: include REF for environmental problems caused by R. kamoji.
-L56-58: this sentence and reference to Figure S2 about a preliminary study that authors have conducted should be deleted from Intro since it is explained later as a result.
-L59: herbicide safeners use and importance should be introduced since they are widely used agrochemicals and enhance herbicide tolerance in cereal crops increasing selectivity in weed control. Explain that there are a wide range of safener chemical classes used for specific crop applications in combination with herbicide partners. Explain why you choose fenclorim among safeners for this study.
-L115: Also 10 seedlings per pot? Indicate.
-L138: It says that all experiments are repeated once…however, table 2 experiment is described in M&M as being repeated twice, and the same for figure 2 for example. Please clarify. Is it because you have pooled the data? you should rewrite this paragraph; it is not clear.
-L172: Why Figure S3 is in supplementary material? You can include it as a regular figure.
-L178-179: Include “each data represents de mean of n experiments….” (as it is written for figure 1).
-L184: Figure 2 is Figure 1. I would like to see a combined figure 1 with supplementary figure S2. In some part of the paper you should include a REF to explain why you choose a-amylase and protein content among other parameters to study the effects of seed germination.
-L194: Table legend indicating what is ED and SI. Tables and figures should be auto-explicative.
-L196-204: I would explain separately both panels of figure 2 to make easier the explanation. In addition, your first sentence is not completely true, not for all doses fenclorim is higher. Describe what you see.
-L210: Eliminate “the same for Figure 3” and write it again the same in figure 3. As I have told you figures should be auto-explicative.
Reviewer 2 Report
Dear Authors,
The manuscript is well written and interesting. I have a few suggestions for the overall manuscript and a very important note.

Round 2
Reviewer 1 Report
Dear Editor and authors,
in my oppinion, the present manuscript deserves to be published, as the experiments are conclusive and show interesting insights into the control of R. kamoji. In fact, the changes included in the second version have improved the quality of the manuscript. However, I just have to ask the authors to correct grammatical errors and English style. For example, there are many simple errors that should have been easily eliminated with the spell checker of the word processing program (singular-plural verbs, butachlor-buatchlor, missing h...). In addition, a correction by a native English speaker would be highly recommended.
Reviewer 2 Report
Dear authors,
Thank you for the revised manuscript. Please find my comments in the attached pdf file.

Author Response
Response to Reviewer 2 Comments
Point 1: Please write subsections with "italics"
Response 1: All the subsections in the manuscript were changed into "italics".
Point 2: Table 1. please write a.i.
Response 2: The unit “ai” was replaced with “a.i.” in Table 1, the abstract and Figure 3.
Point 3: Please write the first letter of each word with Capital letter
Response 3: The words in the “Author Contribution” section was changed to start with Capital letters.